# Influence of Parameter Uncertainty to Stator Current Reconstruction Using Modified Luenberger Observer for Current Sensor Fault-Tolerant Induction Motor Drive

**DOI:** 10.3390/s22249813

**Published:** 2022-12-14

**Authors:** Michal Adamczyk, Teresa Orlowska-Kowalska

**Affiliations:** Department of Electrical Machines, Drives and Measurements, Wroclaw University of Science and Technology, Wyb. Wyspianskiego 27, 50-370 Wroclaw, Poland

**Keywords:** induction motor drive, fault-tolerant control, stator current estimation, modified Luenberger observer

## Abstract

In modern systems with induction motors (IM), in addition to precision, it is also very important to ensure the highest possible reliability and safety. To ensure the above, information about the stator current value is required. If the current sensor (CS) fails, a redundant sensor or an algorithmic solution can be used. The Luenberger observer (LO) can be used to estimate the lost stator current without increasing the cost of the drive system. However, this solution is based on the mathematical model of IM, which is sensitive to its parameters. Therefore, this paper presents a modified LO (MLO) and investigates the effect of a coefficient in the error gain matrix on improving robustness to changes in the IM parameters. As shown by extensive studies, the proposed solution has significantly reduced the influence of the IM parameters on the accuracy of the stator current estimation, which has not been previously reported in the known literature.

## 1. Introduction

Induction motors (IMs) are one of the largest groups of electricity consumers. They are commonly used in drive systems. Therefore, it is very important to ensure their precise control and reliability. This is possible using the vector control methods, among which there are two main ones: direct rotor flux-oriented control (DRFOC) and direct torque control (DTC) [1,2,3]. Both methods require the measurement of phase currents to work properly: in DRFOC for rotor flux estimation and for feedback in the control structure; in DTC for the estimation of the stator flux and the hard-to-measure electromagnetic torque. Therefore, it can be concluded that the stator current is the most important measurable state variable in a drive system with an IM.

Non-invasive LEM type transducers are usually used to determine the current value of the stator current. As the authors note [4], it is a device susceptible to faults, which, due to the design of such a current sensor (CS), can be distinguished: gain change, offset, noise, saturation, open circuit, or intermittent disconnection of circuit [5]. The occurrence of any of these failures will result in the deterioration of the system operation and even loss of stability. Therefore, to minimize the impact of failures on the proper operation of the system, FTC systems should be used, which have been discussed in more detail in [6,7,8].

To realize the CS-FTC system, one of three strategies can be used. The first is based on the DC link current, which was first shown in 1989 [9]. However, this solution requires the use of DC current measurement in the DC link of the voltage source inverter (VSI) and may also result in ripples of current and electromagnetic torque [10]. As a result, this solution is not recommended for systems with a higher degree of security.

The next group of methods is possible only when at least one CS is available. The author in [11] proposes using the asymmetry index to compensate for gain error or measurement offset. However, it should be noted that such a solution would not work with the complete disappearance of the signal from the faulty CS. Detection methods of gain and offset faults have been shown for doubly fed induction generators in [12]. Another solution may be the use of reference signals and various methods to transform phase currents into currents in the reference frame (*α*-*β*) [13,14]. Although the presented methods do not depend on IM parameters, they have two main disadvantages: they are imprecise, especially in transient states, and they cannot be applied when all CSs fail. The authors in [15,16], proposed a CS-FTC structure without a speed sensor. They used a single-phase enhanced phase-locked loop (SEPLL) to reconstruct the faulty phase current without the use of the estimated/measured speed or rotor position. In post-fault operation, the speed and current estimation performance are achieved by the sliding mode observer based on the one measured current and that reconstructed using SEPLL. However, as the authors write, the failure of two CSs requires a switch to the open loop control structure.

The largest group of CS fault compensation methods are methods based on mathematical models. For example, the authors in [17] propose the use of three state observers. In this research, the results have been shown only when at least one CS is available. The authors presented the possibility of detecting fault in one or two of three CSs. One state variable observer with rotor and stator resistance has been shown in [18]. However, the presented solution requires measurement of at least one stator current.

When all CSs are faulty, methods based on flux linkage modeling [19], methods based on the current and voltage model of the rotor flux, called virtual current sensor (VCS) [20,21], as well as state observers, such as an open-loop observer [22] (identical to VCS) or Luenberger observer [23,24,25] can be used to estimate the stator current. However, methods based on mathematical models are more sensitive to changes or misidentification of the IM parameters. Additionally, it should be noted that the IM parameters are not constant and, apart from the temperature, they are also dependent on various operating points. For the types of methods mentioned above, one possibility to minimize the effect of the parameter changes on the quality of the stator current estimation is to use the LO observer; however, this requires the use of an error vector modification as shown in [23].

This paper presents a modified Luenberger observer (MLO) and a detailed analysis of the quality of stator current estimation for various angular speeds and load torques, depending on the changes in parameters of the IM mathematical model. The tests were carried out during both the motoring and regenerating modes. The use of modifications in the calculation of the LO current estimation error allowed for a significant reduction in the influence of the IM parameters on the MLO estimation error, both in the situation when the system worked with two healthy CSs and in the case of post-fault operation (where only one healthy CS was available). Extensive analysis showed even 94% greater robustness of the proposed MLO to changes in IM parameters compared to classical LOs described in the literature, which translates into more precise monitoring of the correct operation of CSs in the drive system, as well as detection and compensation of the fault.

The article consists of five sections. After the introduction, the mathematical model of the classical LO for the estimation of stator current is presented. The third chapter presents the possibility of modifying the stator current estimation error vector in the classical LO, which enables the state error calculation even in a situation when one CS fails. The idea of a modified LO (MLO) consists in the replacement of the measured currents by corrected ones (a combination of measured and estimated phase currents). The next chapter presents an extensive and detailed quality analysis of the stator current estimation using the proposed MLO for various operating conditions (before CS fault and post-fault operation), for motoring and regenerating modes, as well as for different parameter values in the IM model and a wide range of variation of coefficient *k*_0_ in the error gain matrix **G**(*ω_m_*) of the MLO. The tests were carried out for high, medium, and low speeds, as well as for 75% and 25% of the rated torque value. The article ends with conclusions.

## 2. Stator Current Estimation Based on Luenberger Observer

To estimate the stator current, the open loop observer can be used, for which the mathematical model is the same as the model of IM electromagnetic circuits [20]. However, such a solution is very sensitive to changes in IM parameters, as has been demonstrated in the literature [21]. Therefore, as shown in [23], a much better idea is to use the LO. In the classic LO version, an additional matrix **G**(*ω_m_*) multiplied by the current estimation error vector is added to the state equation of IM:(1)$$T_{N}\frac{d}{dt}\hat{x}=A\left(\omega_{m}\right)\hat{x}+Bu_{s}+G\left(\omega_{m}\right)e_{i},$$
with: **x** = col (*i_sα_*, *i_sβ_*, *Ψ_rα_*, *Ψ_rβ_*)—state vector; **u**_s_ = col (*u_sα_*, *u_sβ_*)—input vector; where: *i_sα,β_*—components of the stator current vector **i**_s_; *u_sα,β_*—components of the stator voltage vector **u**_s_; *Ψ_rα,β_*—components of the rotor flux vector **Ψ***_r_*.

The matrices in Equation (1) are defined as follows:– **A**(*ω_m_*) is the state matrix of the IM:
(2)$$A\left(\omega_{m}\right)=\left[\begin{array}{cccc} -\frac{r_{s}}{\sigma l_{s}}-\frac{\left(1-\sigma \right)r_{r}}{\sigma l_{r}} & 0 & \frac{l_{m}r_{r}}{\sigma l_{s}l_{r}^{2}} & \frac{l_{m}}{\sigma l_{s}l_{r}}\omega_{m}\\ 0 & -\frac{r_{s}}{\sigma l_{s}}-\frac{\left(1-\sigma \right)r_{r}}{\sigma l_{r}} & -\frac{l_{m}}{\sigma l_{s}l_{r}}\omega_{m} & \frac{l_{m}r_{r}}{\sigma l_{s}l_{r}^{2}}\\ \frac{l_{m}r_{r}}{l_{r}} & 0 & -\frac{r_{r}}{l_{r}} & -\omega_{m}\\ 0 & \frac{l_{m}r_{r}}{l_{r}} & \omega_{m} & -\frac{r_{r}}{l_{r}} \end{array}\right],$$

– **B** is the input matrix:(3)$$B={\left[\begin{array}{cccc} \frac{1}{\sigma l_{s}} & 0 & 0 & 0\\ 0 & \frac{1}{\sigma l_{s}} & 0 & 0 \end{array}\right]}^{T},$$
where: *r_s_*, *r_r_*—stator and rotor resistances; *l_σs_*, *l_σr_*, *l_m_*—stator and rotor leakage inductances and main inductance; *ω_m_*—rotor speed; *f_sN_*—nominal stator frequency; and *l_r_* = *l_σr_* + *l_m_*, *l_s_* = *l_σs_* + *l_m_*, *σ* = 1 − *l_m_*^2^/(*l_s_l_r_*), *T_N_* = 1/(2π*f_sN_*),

– **e***_i_* is stator current estimation error:


(4)
$$e_{i}=\left[\begin{array}{c} {\hat{i}}_{s\alpha }-i_{s\alpha }\\ {\hat{i}}_{s\beta }-i_{s\beta } \end{array}\right],$$


– **G**(*ω_m_*) is LO gain matrix [26]:(5)$$G={\left[\begin{array}{cccc} g_{1} & g_{2}\omega_{m} & g_{3} & -cg_{2}\omega_{m}\\ -g_{2}\omega_{m} & g_{1} & cg_{2}\omega_{m} & g_{3} \end{array}\right]}^{T},$$
with:(6)$$\begin{array}{l} \begin{array}{cc} g_{1}=-\left(k_{0}-1\right)\left(\frac{r_{s}}{\sigma l_{s}}+\frac{r_{r}}{\sigma l_{r}}\right), & g_{2}=\left(k_{0}-1\right) \end{array},\\ g_{3}=\left(k_{0}^{2}-1\right)\left(\frac{l_{m}r_{r}}{l_{r}}-c\left(\frac{r_{s}}{\sigma l_{s}}+\frac{\left(1-\sigma \right)r_{r}}{\sigma l_{r}}\right)\right)-cg_{1}, \end{array}$$
and *c* = *σl_s_l_r_*/*l_m_*, *k*_0_ is a positive coefficient.

The mathematical model of LO above is presented in the p.u. system [3], using the spatial vectors of the IM written in the stationary reference frame *α-β*.

## 3. Modification of the Classical Luenberger Observer

In the case of a CS failure, the observer error **e***_i_* is incorrectly counted, resulting in incorrect operation of the observer and the entire drive system. Therefore, a modified LO (MLO) was used in the present study. In the stator current error vector, the measured currents, **i***_s_*, are replaced by corrected currents, **i***_s_^c^*:(7)$$e_{i}=\left[\begin{array}{c} {\hat{i}}_{s\alpha }-i_{s\alpha }^{c}\\ {\hat{i}}_{s\beta }-i_{s\beta }^{c} \end{array}\right],$$
which is the combination of the phase currents measured by healthy CSs, **i***_s,_* and the currents ${\hat{i}}_{s}$ estimated by the LO, for four different scenarios:(8)$$\left[\begin{array}{c} i_{s\alpha }^{c}\\ i_{s\beta }^{c} \end{array}\right]=\{\begin{array}{ccc} \left[\begin{array}{c} i_{sA}\\ \frac{\sqrt{3}}{3}\left(i_{sA}+2i_{sB}\right) \end{array}\right] & \mathrm{for} & \mathrm{both}\;\mathrm{healthy}\;\mathrm{CS},\\ \left[\begin{array}{c} -i_{sB}-{\hat{i}}_{sC}\\ \frac{\sqrt{3}}{3}\left({\hat{i}}_{sA}+2i_{sB}\right) \end{array}\right] & \mathrm{for} & \mathrm{faulty}\;\mathrm{CS}\;\mathrm{in}\;\mathrm{phase}\;A,\\ \left[\begin{array}{c} i_{sA}\\ \frac{\sqrt{3}}{3}\left(i_{sA}+2{\hat{i}}_{sB}\right) \end{array}\right] & \mathrm{for} & \mathrm{faulty}\;\mathrm{CS}\;\mathrm{in}\;\mathrm{phase}\;B,\\ \left[\begin{array}{c} {\hat{i}}_{s\alpha }\\ {\hat{i}}_{s\beta } \end{array}\right] & \mathrm{for} & \mathrm{both}\;\mathrm{faulty}\;\mathrm{CS}. \end{array}$$

When one CS is available, the required estimated phase currents (${\hat{i}}_{sA},\;{\hat{i}}_{sB},\;{\hat{i}}_{sC}$) in Equation (8) can be calculated using the inverse Clarke transform of the stator current components estimated by LO:(9)$$\left[\begin{array}{c} {\hat{i}}_{sA}\\ {\hat{i}}_{sB}\\ {\hat{i}}_{sC} \end{array}\right]=\frac{1}{2}\left[\begin{array}{c} 2{\hat{i}}_{s\alpha }\\ -{\hat{i}}_{s\alpha }+\sqrt{3}{\hat{i}}_{s\beta }\\ -{\hat{i}}_{s\alpha }-\sqrt{3}{\hat{i}}_{s\beta } \end{array}\right].$$

It should be mentioned that the presented solution will also work in the case of failure of all current sensors. However, the estimation error cannot then be calculated because the vector **e***_i_* is equal to zero.

## 4. Dependence of the Stator Current Estimator Quality to Changes in the IM Parameters

As has been shown in many works on LOs for IM drives, the quality of the estimation of the state variables depends on the value of the design coefficient *k_0_* used in the gain matrix **G** of the observer mathematical model (1). The correct choice of this parameter influences not only the observer dynamics but also its sensitivity to changes in motor parameters [25,26,27]. In the case of CS failures this influence can even be greatest. Therefore, in this article we tested the influence of coefficient *k*_0_ on the quality of the stator current estimation for changes in the IM parameters and in different failures: two healthy CSs, faulty CS in phase *A*, and faulty CS in phase *B*. The tests were carried out in both the motoring and regenerating modes. The range of speed and torque variations was as follows: *ω_m_^ref^* ∈ {±0.1, ±0.5, ±1.0}*ω_mN_*, *t_L_^ref^* ∈ {0.25, 0.75}*t_LN_* and for wide range of *k*_0_ coefficient: *k*_0_∈{0.6, 1.0, 1.4, 1.8, 2.2, 2.6, 3.0}. Three IM parameters were considered in the study: rotor resistance *r_r_*, stator resistance *r_s_*, and main inductance *l_m_*. The values of the mentioned parameters were increased by 50% in the IM model under simulation studies.

All tests have been carried out for the DRFOC structure, with feedback from measured stator currents, for the IM parameters shown in Appendix A (Table A1). The quality of the estimation of the stator current in Phase *A* or *B* has been checked based on the comparison of the stator currents measured in the drive system and calculated by the MLO in different faulty modes, using the root mean square error (RMSE), for the time range *t*_2_ − *t*_1_ = 1s and for each *k*th sample from *t*_1_/*T_s_* to *t*_2_/*T_s_*, where *T_s_* is the sampling time:(10)$$\Delta i_{sA/B}=\sqrt{\frac{\sum_{k=t_{1}/T_{s}}^{t_{2}/T_{s}}{\left(i_{sA/B}\left(k\right)-{\hat{i}}_{sA/B}\left(k\right)\right)}^{2}}{\left(t_{2}-t_{1}\right)/T_{s}+1}}.$$

### 4.1. Healthy CSs

In this case, the quality of stator current estimation has been checked for two healthy CSs. RMSE values in phases *A* and *B* have been shown for *t_L_^ref^* = 0.25*t_LN_* in Figure 1 and for *t_L_^ref^* = 0.75*t_LN_* in Figure 2. The values from minimum to maximum were sorted using a colour gradient. Additionally, the lowest value at a given speed was marked in orange and values bigger then maximum of *z*-axis are marked in light purple.

As can be seen in Figure 1, for low load, the rotor resistance and the main inductance have the greatest influence on the quality of the stator current estimation. For low angular velocity range, the biggest impact of rotor resistance can be observed (Figure 1c,d).

For 75% of the rated load (Figure 2), the rotor resistance also has a big impact on the stator current estimation; however, the effect of the main inductance is less than for 25% of the rated load torque. The larger error has been obtained for low speeds and changing stator resistance.

For the results obtained, Table 1 was prepared in which the number of smallest RMSE values for individual *k*_0_ was chosen according to Figure 1 and Figure 2. As can be seen, the best quality of stator current estimation has been obtained the most often for *k*_0_ = 3.0 (66 of 72 cases). Therefore, this value can be recommended for stator current estimation using the MLO (e.g., to make the fault detector robust to changes in IM parameters) for two healthy CSs.

When the *k*_0_ is different than 1, the percentage decrease in the sensitivity of the MLO to changes in IM parameters relative to the open loop observer—OLO (MLO with *k*_0_ = 1) was checked using the following expression:(11)$$\delta i_{sA/B}=\frac{\Delta i_{sA/B}^{k_{0}=1.0}-\Delta i_{sA/B}^{k_{0}\neq 1.0}}{\Delta i_{sA/B}^{k_{0}=1.0}}\cdot 100\%.$$

Because the IM drive mainly operates around the rated load, the results are shown for *t_L_^ref^* = 0.75*t_LN_* (Figure 2). Since the RMSE results are calculated for changes of three IM parameters, the relative error $\delta i_{sA/B}$ was calculated as the average value for all of them. The results obtained are shown in Table 2 with the best improvements for different speed values, highlighted in gray color.

As can be seen in Table 2, for *k*_0_ = 3.0, the mean decrease in the impact of the IM parameters is mostly between 75% and 94%, except for low speed in the motoring mode (41%). With two healthy CSs, the *k*_0_ = 0.6 can result in up to twice the estimation error.

### 4.2. Faulty CS in Phase A

In this study, a faulty current sensor in phase *A* was assumed. Therefore, the corrected currents from Equation (8) used in the observer feedback are calculated according to this situation. The effect of the parameters for 25% of the rated load is shown in Figure 3 and for 75% in Figure 4.

As can be seen in Table 3, which presents the number of smallest RMSE values for individual *k*_0_ when CS in phase *A* is faulty, the quality of the stator current estimation in phase *A* does not show the best effect for a particular value of *k*_0_. It should be noted that for *k*_0_ = 3.0, the largest error in current estimation can be observed with increased main inductance or stator resistance and low load. The main inductance increases naturally when the field is weakened, while the stator resistance has the greatest effect at low speeds, so *k*_0_ = 3.0 can be recommended. For the current in phase *B*, the best estimation quality was definitely obtained for *k*_0_ = 3.0 for both load torque values: in 34 of 36 cases (Figure 3b,d,f and Figure 4b,d,f).

According to Equation (11) the percentage decrease in MLO sensitivity in relation to OLO has been checked. The results obtained are shown in Table 4.

As can be seen, under the low load torque, the estimation quality of the missing current (in phase *A*) is much worse than that for the available current. It is worth noting, however, that drive systems are designed to operate around rated loads, while an average improvement of about 23% is already satisfactory. Additionally, the situation that is unravelling concerns post-fault operations. It is recommended that this solution be used only in CS-FTC systems. Estimation of the available current (*i_sB_*) shows between 56% and almost 93% quality improvement. Thus, this solution can find wide applications in residuum-based fault detectors.

### 4.3. Faulty CS in Phase B

Similarly, to the previous subsections, the quality of current estimation in the case of CS failure in phase *B* was analyzed for a wide range of speed and *k*_0_ coefficient variations. The results obtained for 25% and 75% of the rated load are shown in Figure 5 and Figure 6, respectively. For *k*_0_ = 3, the system lost stability at some operation conditions, as highlighted by the red line. The numerical value of the smallest RMSE for specific *k*_0_ is shown in Table 5, while the percentage improvement is shown in Table 6.

As can be seen in Figure 5 and Figure 6 and Table 5, the best estimate quality of the *A*-phase stator current was obtained most often for *k*_0_ = 3.0 (in 28 out of 36 cases). However, it should be noted that for this value of *k*_0_, the system loses stability at some points of operation (highlighted X in Table 6), so it is recommended to use *k*_0_ = 2.6, for which the improvement is also very large (Table 6). On the other hand, the lowest RMSE values for the estimation of the stator current in phase *B* can be seen for *k*_0_ = 0.6 (in 22 of 36 cases). Therefore, such values of *k*_0_ are recommended in the case of a faulty CS in phase *B*.

As can be seen in Table 6 for the available CS in phase *A*, the current estimation in this phase is from 35% to 73% and is more accurate for 75% of the nominal load (for *k*_0_ = 2.6). For phase *B*, the quality improvement in the current estimation is about 10−43%. It is worth noting that significant improvements can be seen, especially in the regenerating mode.

## 5. Conclusions

In the known literature, in the case of LOs, the authors take *k*_0_ close to 1 for the estimation of the stator current. It is worth noting, however, that the application of the LO modification allows us to use a different value of *k*_0_, due to the correct way of calculating the estimation error, which is justified based on the analysis of the obtained results.

For current estimation in a phase where a healthy CS is available, it is recommended to use *k*_0_ between 2.6 and 3.0. This increases the robustness to changes in IM parameters by up to 93% compared to OLOs, or LOs with *k*_0_ = 1, as used in the literature. This makes it possible to design accurate and robust CS fault detectors. When the CS in phase *A* is corrupted, the best quality of estimation was obtained for *k*_0_ between 2.6 and 3.0. However, when the CS in phase *B* is faulty, *k*_0_ = 0.6 is recommended.

It is worth mentioning that the presented MLO solution can be successfully applied to CS fault-tolerant systems, both when designing a high-quality detector and for fault compensation. Furthermore, the proposed solution allows the detection and compensation of all CSs; however, the estimation error information cannot be obtained in this situation, so the solution reduces to OLO. Nevertheless, it should be emphasized that the vector control in post-fault mode is only to ensure safe operation of the drive system, and a faulty CS should be repaired or replaced as soon as possible.

## Figures and Tables

**Figure 1 sensors-22-09813-f001:**
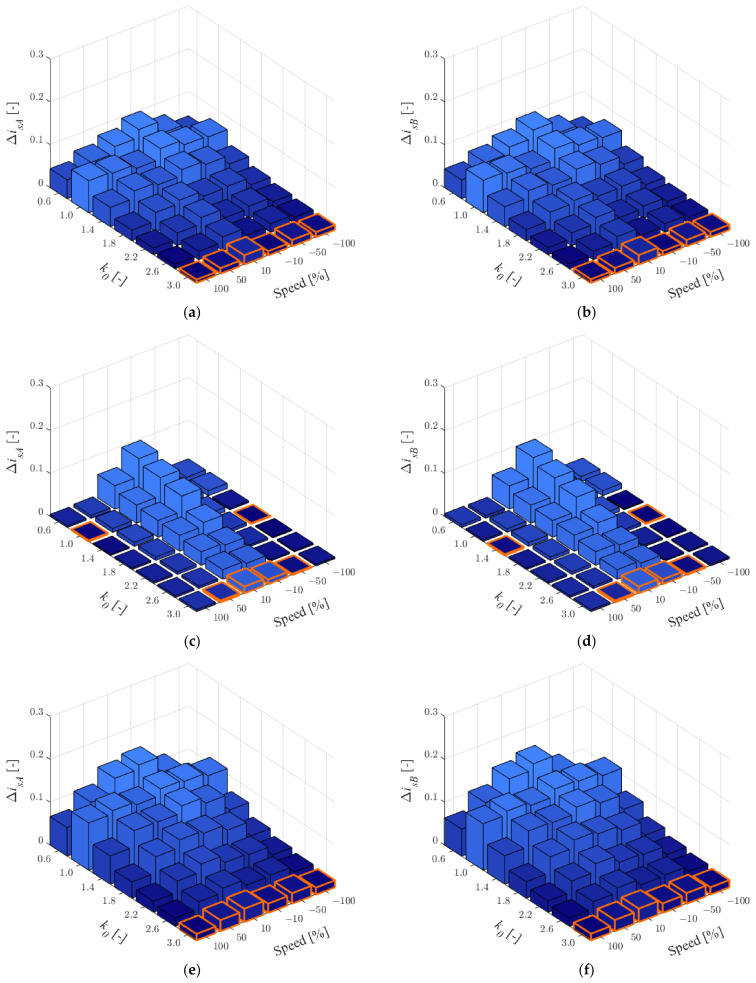
RMSE values for stator currents in phase *A* (**a**,**c**,**e**) and *B* (**b**,**d**,**f**) for *r_r_^IM^* = 1.5*r_rN_* (**a**,**b**), *r_s_^IM^* = 1.5*r_sN_* (**c**,**d**), *l_m_^IM^* = 1.5*l_mN_* (**e**,**f**) and *t_L_^ref^* = 0.25*t_LN_*, when both CSs are healthy.

**Figure 2 sensors-22-09813-f002:**
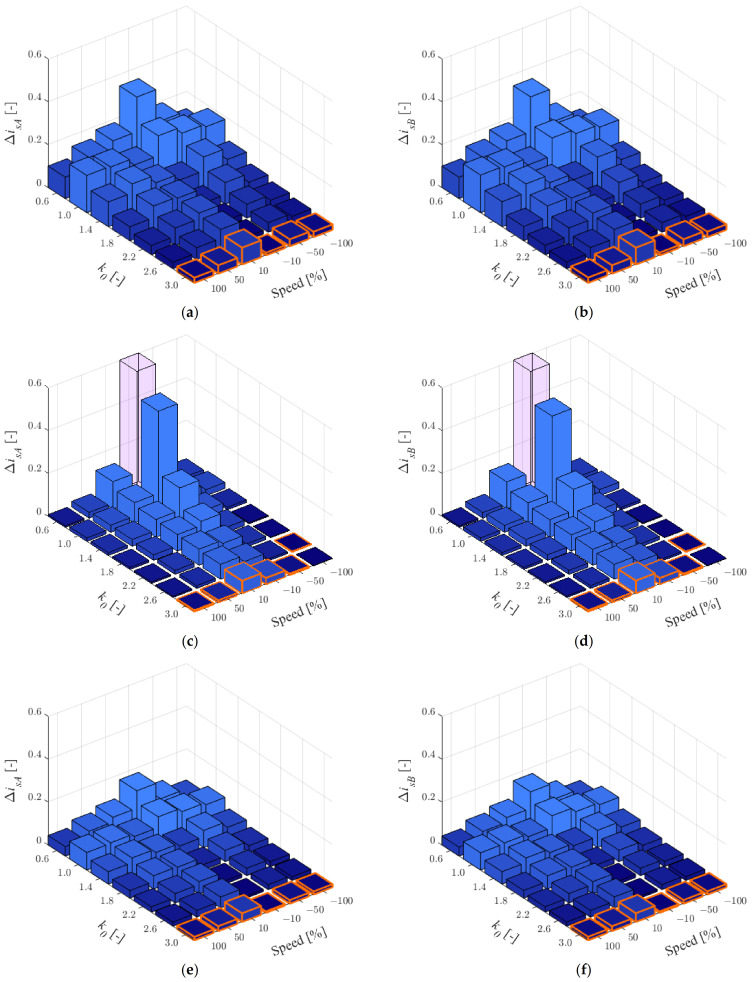
RMSE values for stator currents in phase *A* (**a**,**c**,**e**) and *B* (**b**,**d**,**f**) for *r_r_^IM^* = 1.5*r_rN_* (**a**,**b**), *r_s_^IM^* = 1.5*r_sN_* (**c**,**d**), *l_m_^IM^* = 1.5*l_mN_* (**e**,**f**) and *t_L_^ref^* = 0.75*t_LN_*, when both CSs are healthy.

**Figure 3 sensors-22-09813-f003:**
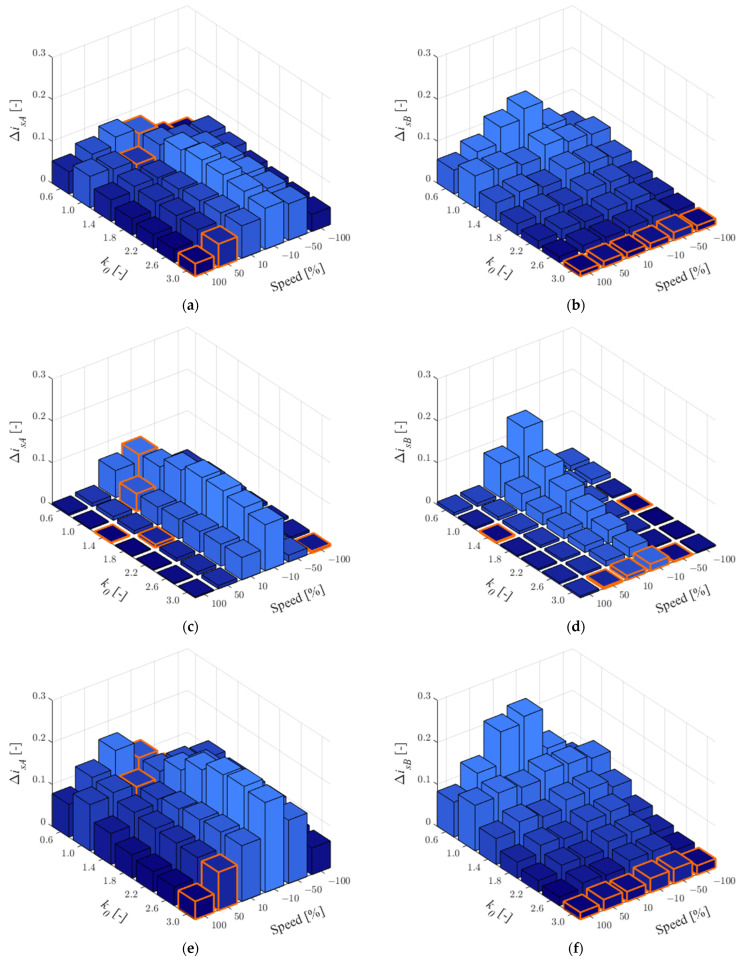
RMSE values for stator currents in phase *A* (**a**,**c**,**e**) and *B* (**b**,**d**,**f**) for *r_r_^IM^* = 1.5*r_rN_* (**a**,**b**), *r_s_^IM^* = 1.5*r_sN_* (**c**,**d**), *l_m_^IM^* = 1.5*l_mN_* (**e**,**f**) and *t_L_^ref^* = 0.25*t_LN_*, when the CS in phase *A* is faulty.

**Figure 4 sensors-22-09813-f004:**
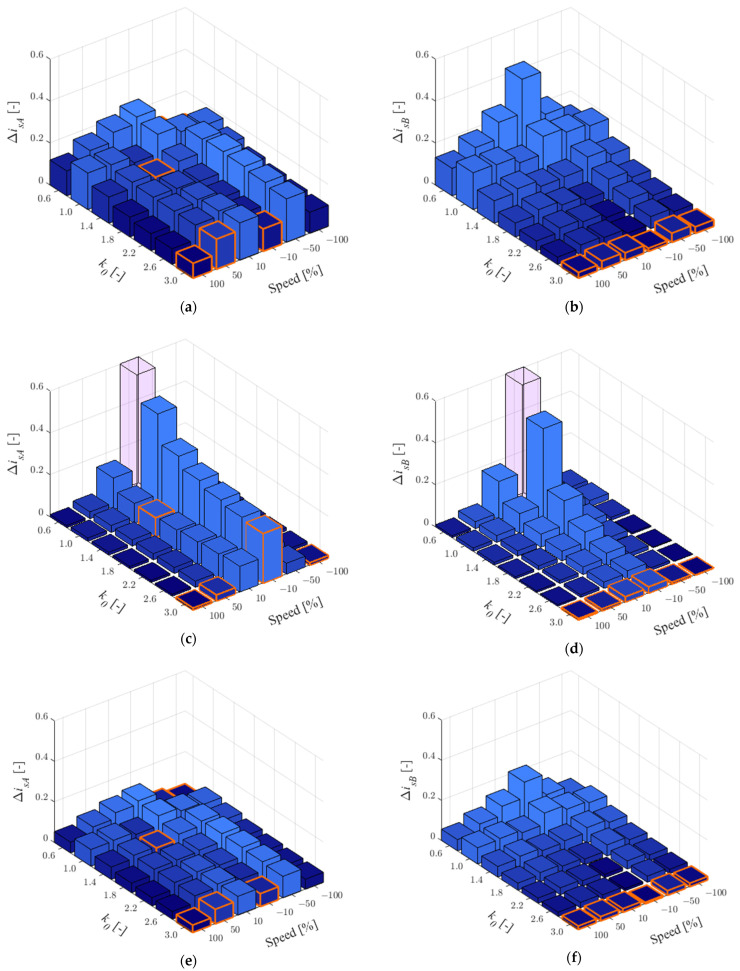
RMSE values for stator currents in phase *A* (**a**,**c**,**e**) and *B* (**b**,**d**,**f**) for *r_r_^IM^* = 1.5*r_rN_* (**a**,**b**), *r_s_^IM^* = 1.5*r_sN_* (**c**,**d**), *l_m_^IM^* = 1.5*l_mN_* (**e**,**f**) and *t_L_^ref^* = 0.75*t_LN_*, when the CS in phase *A* is faulty.

**Figure 5 sensors-22-09813-f005:**
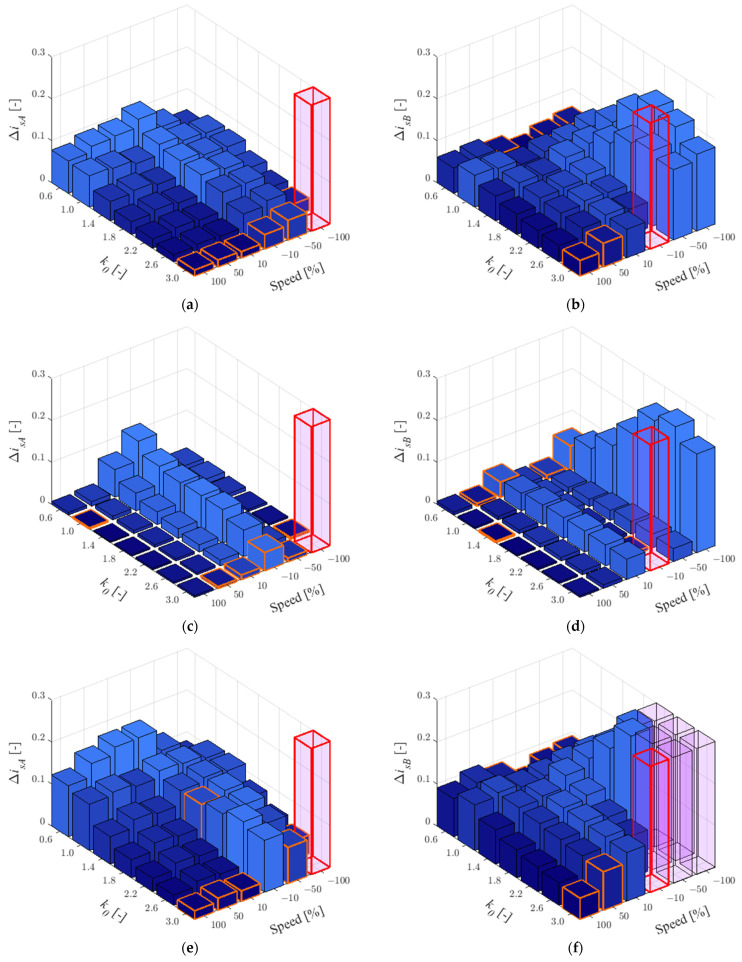
RMSE values for stator currents in phase *A* (**a**,**c**,**e**) and *B* (**b**,**d**,**f**) for *r_r_^IM^* = 1.5*r_rN_* (**a**,**b**), *r_s_^IM^* = 1.5*r_sN_* (**c**,**d**), *l_m_^IM^* = 1.5*l_mN_* (**e**,**f**) and *t_L_^ref^* = 0.25*t_LN_*, when the CS in phase *B* is faulty.

**Figure 6 sensors-22-09813-f006:**
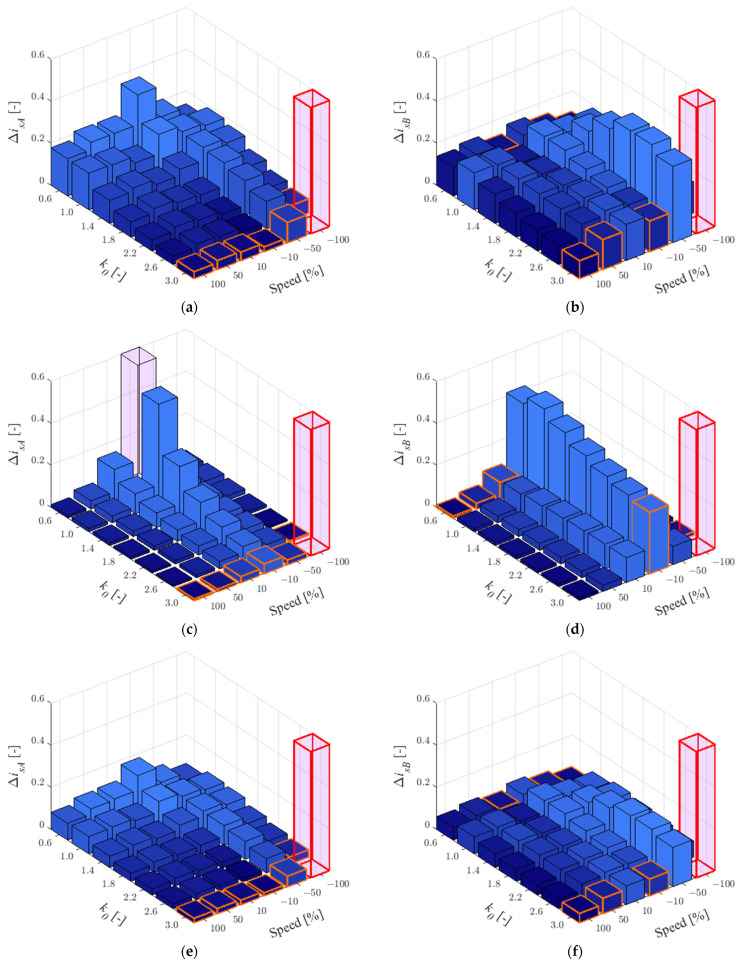
RMSE values for stator currents in phase *A* (**a**,**c**,**e**) and *B* (**b**,**d**,**f**) for *r_r_^IM^* = 1.5*r_rN_* (**a**,**b**), *r_s_^IM^* = 1.5*r_sN_* (**c**,**d**), *l_m_^IM^* = 1.5*l_mN_* (**e**,**f**) and *t_L_^ref^* = 0.75*t_LN_*, when the CS in phase *B* is faulty.

**Table 1 sensors-22-09813-t001:** The number of smallest RMSE values for individual *k*_0_.

	25% of *t_LN_*						75% of *t_LN_*					
	$\Delta i_{sA}$			$\Delta i_{sB}$			$\Delta i_{sA}$			$\Delta i_{sB}$		
*k_0_*	*r_r_*	*r_s_*	*l_m_*	*r_r_*	*r_s_*	*l_m_*	*r_r_*	*r_s_*	*l_m_*	*r_r_*	*r_s_*	*l_m_*
0.6	0	0	0	0	0	0	0	0	0	0	0	0
1.0	0	1	0	0	0	0	0	0	0	0	0	0
1.4	0	0	0	0	1	0	0	0	0	0	0	0
1.8	0	1	0	0	1	0	0	0	0	0	0	0
2.2	0	0	0	0	0	0	0	0	0	0	0	0
2.6	0	0	0	0	0	0	0	1	0	0	1	0
3.0	6	4	6	6	4	6	6	5	6	6	5	6

**Table 2 sensors-22-09813-t002:** Decrease in MLO sensitivity to changes in IM parameters relative to OLO, for two healthy CSs (grey color indicates the best improvement).

	$\delta i_{sA}\;[\%]$						$\delta i_{sB}\;[\%]$					
*k_0_* [-]/Speed [%]	−100	−50	−10	10	50	100	−100	−50	−10	10	50	100
0.6	35.5	4.0	−93.1	−34.0	13.0	44.1	35.7	3.7	−104.7	−34.0	13.5	45.4
1.4	46.6	23.1	60.6	13.0	6.9	31.3	46.5	22.9	59.7	13.0	6.7	30.4
1.8	68.7	50.1	79.9	17.8	28.2	59.6	68.5	49.7	79.3	17.6	27.9	58.6
2.2	78.5	67.8	87.9	20.8	49.7	72.2	78.4	67.6	87.5	20.3	49.3	71.4
2.6	83.7	78.2	91.9	28.1	65.1	78.9	83.5	78.0	91.6	27.5	64.8	78.3
3.0	86.7	84.4	94.2	41.7	75.2	83.1	86.3	84.3	94.0	41.1	74.9	82.6

**Table 3 sensors-22-09813-t003:** The number of smallest RMSE values for individual *k*_0_.

	25% of *t_LN_*						75% of *t_LN_*					
	$\Delta i_{sA}$			$\Delta i_{sB}$			$\Delta i_{sA}$			$\Delta i_{sB}$		
*k_0_*	*r_r_*	*r_s_*	*l_m_*	*r_r_*	*r_s_*	*l_m_*	*r_r_*	*r_s_*	*l_m_*	*r_r_*	*r_s_*	*l_m_*
0.6	3	2	3	0	0	0	2	1	2	0	0	0
1.0	1	1	1	0	0	0	0	0	0	0	0	0
1.4	0	1	0	0	1	0	1	1	1	0	0	0
1.8	0	1	0	0	1	0	0	0	0	0	0	0
2.2	0	0	0	0	0	0	0	0	0	0	0	0
2.6	0	0	0	0	0	0	0	0	0	0	0	0
3.0	2	0	2	6	4	6	3	4	3	6	6	6

**Table 4 sensors-22-09813-t004:** Decrease in MLO sensitivity to changes in IM parameters relative to OLO, for a faulty CS in phase *A* (grey color indicates the best improvement).

	$\delta i_{sA}\;[\%]$						$\delta i_{sB}\;[\%]$					
*k_0_* [-]/Speed [%]	−100	−50	−10	10	50	100	−100	−50	−10	10	50	100
0.6	51.2	39.1	−36.2	−57.9	−1.1	32.9	26.9	−9.5	−139.4	−92.9	−8.6	28.0
1.4	14.5	−14.5	27.0	2.3	10.0	26.7	38.7	27.0	59.6	34.0	27.6	36.4
1.8	30.3	−17.4	38.2	−5.5	13.4	43.7	61.9	47.5	78.5	51.0	47.0	58.9
2.2	38.6	−14.7	44.4	−14.6	13.9	51.1	73.3	61.6	86.7	61.3	59.8	70.1
2.6	43.5	−9.3	48.3	−22.7	14.2	54.8	79.8	71.5	90.9	68.3	68.7	76.6
3.0	47.0	−2.8	51.0	−29.0	15.1	56.8	83.7	78.4	93.4	73.6	75.1	80.8

**Table 5 sensors-22-09813-t005:** The number of smallest RMSE values for individual *k*_0_.

	25% of *t_LN_*						75% of *t_LN_*					
	$\Delta i_{sA}$			$\Delta i_{sB}$			$\Delta i_{sA}$			$\Delta i_{sB}$		
*k_0_*	*r_r_*	*r_s_*	*l_m_*	*r_r_*	*r_s_*	*l_m_*	*r_r_*	*r_s_*	*l_m_*	*r_r_*	*r_s_*	*l_m_*
0.6	0	0	0	4	4	4	0	0	0	3	4	3
1.0	0	1	0	0	0	0	0	0	0	0	0	0
1.4	0	0	0	0	1	0	0	0	0	0	0	0
1.8	0	0	1	0	0	0	0	0	0	0	0	0
2.2	0	0	0	0	0	0	0	0	0	0	0	0
2.6	1	1	1	0	1	0	1	1	1	0	1	0
3.0	5	4	4	2	0	2	5	5	5	3	1	3

**Table 6 sensors-22-09813-t006:** Decrease in MLO sensitivity to changes in IM parameters relative to OLO, for a faulty CS in phase *B* (grey color indicates the best improvement).

	$\delta i_{sA}\;[\%]$						$\delta i_{sB}\;[\%]$					
*k*_0_ [-]/Speed [%]	−100	−50	−10	10	50	100	−100	−50	−10	10	50	100
0.6	25.2	−2.8	−73.3	−54.0	−24.0	1.2	43.8	30.4	17.1	10.5	10.6	24.7
1.4	9.8	4.3	45.6	27.0	25.2	35.0	−15.1	−34.4	5.6	−9.6	0.2	18.9
1.8	36.3	8.9	68.7	43.8	43.6	55.2	−0.8	−72.3	16.5	−17.1	2.9	32.4
2.2	54.6	18.9	80.2	55.4	56.7	66.4	12.9	−99.3	25.1	−22.0	6.6	40.2
2.6	65.9	35.8	86.5	64.1	66.2	73.3	22.5	−99.1	31.2	−24.4	10.7	45.1
3.0	X	52.8	90.2	70.7	73.1	78.0	X	−80.0	35.7	−25.0	14.9	48.5

## Data Availability

Not applicable.

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
