# Peer review of "Influence of Parameter Uncertainty to Stator Current Reconstruction Using Modified Luenberger Observer for Current Sensor Fault-Tolerant Induction Motor Drive"

_sensors, 2022, doi:10.3390/s22249813_

Round 1

Reviewer 1 Report

1On page 4, there is no Table A1 in the text, please check;

2The physical meaning of the symbols in the text and tables should be clear, such as formula (8) and formula (10);

3In the part of experimental analysis, it is suggested to give the comparison results with some existing methods to verify the necessity and innovation of this work.

Author Response

Dear Reviewer,

thank you very much for reviewing our manuscript. The attached file contains the answers to your questions.

Yours sincerely
Authors

Reviewer 2 Report

Dear Authors,

 This paper examines the effect of parameter mismatches over the stator current reconstruction method based on the Luenberger observer (LO) for the current-sensor fault-tolerant induction motor drive and proposed a gain selection method. After a detailed review, the points to consider are listed as follows:

   - The manuscript is generally well written but the literature review should be improved by citing other stator current reconstruction methods (model-based or non-model-based) and by discussing the pros and cons between them.

   - In this manuscript, a gain selection method has been proposed taking into account the parameter sensitivity, instead of a modified LO. As far as I can see, the manuscript contains a standard LO mechanism, and there is no modified LO.

   - Another important problem with the manuscript is the lack of experimental validation. It would be better to verify the simulation results through the experiments.

   - I think the k_0 range is not enough. All tests show that higher k_0 values (in the selected range) provide less parameter sensitivity. The question is: Does a value of 2,4 for k_0 provide less parameter sensitivity?

   - The manuscript does not provide a generic selection methodology. It simply analyzes the effect of k_0 under healthy and faulty operations of current sensors. This rasps the main contribution.

Considering the above, I think that the manuscript is not ready for publication and should be revised in line with the comments.

Author Response

(The authors gave the same response as above.)

Round 2

Reviewer 2 Report

Dear Authors,

Thank you for your response to my comments. All have been taken into account in the revised manuscript. It was important to extend the k_0 range.